# *Pseudomonas* sp., Strain L5B5: A Genomic and Transcriptomic Insight into an Airborne Mine Bacterium

Jose Luis Gonzalez-Pimentel [1], Irene Dominguez-Moñino [2], Valme Jurado [2], Ana Teresa Caldeira [1] and Cesareo Saiz-Jimenez [2],*

1 Laboratorio Hercules, Universidade de Evora, 7004-516 Evora, Portugal
2 Instituto de Recursos Naturales y Agrobiologia, IRNAS-CSIC, 41012 Sevilla, Spain
* Correspondence: saiz@irnase.csic.es

**Abstract:** Mines, like other subterranean environments, have ecological conditions which allow the thriving of microorganisms. Prokaryotes and fungi are common inhabitants of mines, developing a metabolism suitable for growing in such inhospitable environments. The mine of Lousal, Portugal, is an interesting site for the study of microorganisms present in their galleries. Aerobiological studies resulted in the isolation of a *Pseudomonas* sp., strain L5B5, closely related to the opportunistic fish pathogen *P. piscis* MC042T, and to the soil bacteria *P. protegens* CHA0T, *P. protegens* Cab57, and *P. protegens* Pf-5. Strain L5B5 was able to inhibit the growth of the pathogenic bacteria *Bacillus cereus*, *Staphylococcus aureus*, and *Acinetobacter baumanii*, as well as the cave fungi *Aspergillus versicolor*, *Penicillium chrysogenum*, *Cladosporium cladosporioides*, *Fusarium solani*, and *Ochroconis lascauxensis*. In silico analyses based on de novo genome hybrid assembly and RNA-Seq, performing seven conditions based on culture and phases of growth resulted in the prediction and detection of genetic mechanisms involved in secondary metabolites, with the presence of a possible new gene cluster transcribed under the tested conditions, as well as feasible virulence factors and antimicrobial resistance mechanisms.

**Keywords:** *Pseudomonas*; airborne bacteria; bioactive compounds; Lousal mine; predicted gene clusters

---

## 1. Introduction

Mines shape ecosystems that harbor oligotrophic microorganisms adapted to an extreme environment. In most cases, microbes have evolved mechanisms to resist high heavy metal concentrations [1,2]. Microbial communities in mines are largely represented by *Proteobacteria*, followed by *Acidobacteria*, *Firmicutes*, and *Bacteroidetes* [3]. Among the *Proteobacteria*, the genus *Pseudomonas* is commonly retrieved in mines [2,4], as well as in other subterranean environments such as caves [5–7].

Lousal mine (Azinheira dos Barros, Portugal) is located in the Iberian Pyrite Belt, one of the largest massive sulfide deposits in the world. The mine was active from 1900 to 1988 and exploited mainly for pyrite. However, mining was abandoned due to the low content of zinc and copper in the ores, making the activity not viable economically. After the mine closure, the rather well-preserved surface infrastructures were adapted for a Mining Museum and a center for education in science and technology. The neighboring contaminated area was rehabilitated, and a program of geo-tourism was launched [8]. In the old Lousal mine, the Waldemar Gallery is open to visitors and a sampling campaign was carried out for isolating airborne bacteria.

Aerobiological studies in caves were recently adopted as a mean to forecast and control microbial outbreaks [9,10]. The data showed that species of *Pseudomonas* were commonly isolated from the air [11], but studies on airborne bacteria are infrequent in mines. There, we isolated an interesting *Pseudomonas* sp., strain L5B5, in Lousal mine.

*Pseudomonas* is a versatile genus of bacteria widely found in environments such as soils and waters, as well as involved in cooperative relations or, in contrast to beneficial interactions, causing infections in other organisms [12]. Well known cases of beneficial bacteria are

---

some strains of *P. fluorescens* and *P. protegens* [13,14], whereas *P. syringae* or the extensively studied opportunistic human pathogen *P. aeruginosa* could represent faithfully the best examples of dangerous members in this genus for plants and humans, respectively [15].

*Pseudomonas* includes a group of bacteria synthesizing valuable secondary metabolites for medicine and biotechnology. For instance, the polyketide antibiotic mupirocin produced by *P. fluorescens* NCIMB 10586 was originally isolated in 1971 and, it is still used to control infection of *Staphylococcus aureus* on the skin and nasal cavity [16]. Likewise, phenazines stand out among other bacterial secondary metabolites, being one of the most abundant and the most studied compounds. Phenazines-producing bacteria are usually associated with host organisms, presenting antagonistic roles depending on the related organism. This is the case of pyocyanin, produced by *P. aeruginosa*, which was originally isolated from human wounds and, in consequence, linked to diseases. However, phenazines synthesized by fluorescent plant pseudomonads were described as a beneficial activity for plant growth and pathogens protection [17]. More recently, during the first decade of the 21st century, a biosurfactant compound, orfamide, was discovered in *P. protegens* Pf-5 [18]. Afterwards, different metabolites were produced by some *Pseudomonas*, and their antimicrobial activities described [19].

Siderophores are also very common secondary metabolites in *Pseudomonas.* Iron is an essential element for many redox reactions in bacteria, but since this metal is not bioavailable under aerobic conditions, siderophores represent the best strategy that microorganisms have developed to acquire this element. Pyochelin and pyoverdine are the two most usual siderophores in *Pseudomonas* to chelate ferric iron and release it in the cytoplasm under iron-limiting conditions. Siderophores were extensively studied in *P. aeruginosa*, due to their involvement in virulence factors [20]. Similar to phenazines, the function of siderophores in non-pathogenic *Pseudomonas* appears to be exclusively addressed to promote growth under iron limitation [21].

In addition, some *Pseudomonas* have the consideration of biocontrol agents not only because of siderophores production, which favors the absence of pathogens of plants, but also due to their capabilities to synthesize antifungals, such as pyrrolnitrin and pyoluteorin [22,23]. Here, we report the in vitro and in silico analyses of *Pseudomonas* sp., strain L5B5, isolated from the air of a pyrite mine in Lousal, Portugal.

## 2. Materials and Methods

### 2.1. Sampling, Isolation, and Identification of the Bacterium

Air samples from the mine were collecting using a Surface Air System (Duo SAS, model Super 360, International pBI, Milan, Italy). Samples were taken in duplicate, and the volume of filtered air was set to 100 L as reported by Martin-Sanchez et al. [10]. The culture medium was Trypticase-soy-agar (TSA) and plates were incubated at 28 °C. Morphologically different colonies were isolated from TSA culture medium. Bacterial identification was carried out by sequencing the 16S ribosomal RNA (rRNA) gene [11]. A *Pseudomonas* sp., strain L5B5, deposited at the IRNAS-CSIC collection, was of interest and subjected to further studies.

### 2.2. Bacterial Inhibition Assays

Two different assays were used. Strain L5B5 was cultured in nutrient agar with 2% of glycerol (NA-glycerol) to promote the expression of genes involved in the secondary metabolism [24]. Plates were incubated at 28 °C for 24 h. After that, 0.4 cm diameter of biomass was removed from culture with a hole puncher and stuck in TSA plates with each pathogenic bacterium, which were prepared as follows: The pathogen bacterium was cultured in 7 mL of trypticase soy broth (TSB) at 30 °C and 180 rpm between 18–24 h until reached a McFarland standard 2. This culture was added to a 30 mL sterilized TSA close to jellify (45–50 °C). Once plates solidified, 0.4 cm biomass of the tested bacteria was stuck to the plates and cultured for 24 h at 30 °C. Inhibition of pathogens was successful when zone of inhibition around the L5B5 strain biomass was observed. Bioactivity was tested

against both Gram-positive and Gram-negative bacteria, including *Bacillus cereus* CECT 148, *Staphylococcus aureus* CECT 4630, *Escherichia coli* DSM 105182, *Pseudomonas aeruginosa* CECT 110, and *Acinetobacter baumannii* DSM 30007. These pathogenic bacteria were used for both first and second assays.

In the second assay, strain L5B5 was cultured in a flask with 150 mL of nutrient broth (NB) with 2% of glycerol (*v/v*) at 30 °C and 180 rpm until saturation was reached. The same procedure was carried out using glucose, mannose and fructose as alternative carbon source aiming to determine the differential gene expression, according to the culture conditions. After incubation, the cultures were filtered through a nucleopore polycarbonate filter with 0.22 μm pore size and the filtrate was tested for antibacterial activity. Pathogenic bacterial suspensions were prepared in 0.9% (*w/v*) sterile saline solution and adjusted to 0.5 McFarland units using a densitometer BioSan DEN-1B with wavelength λ = 565 ± 15 nm (BioSan, Riga, Latvia). Inhibitory analysis was carried out in duplicate in a SPECTROstarNano microplate's device (BMG Labtech, Offenburg, Germany) adding a volume of 150 μL to each well. The wells contained 20 μL pathogenic saline solution + 30 μL culture medium + 100 μL filtrate, a negative control with 100 μL filtrate + 50 μL of culture medium, a positive control (20 μL pathogenic saline solution + 130 μL culture medium), and a blank well (150 μL culture medium). The microplate was set up as follows: orbital shaking at 500 rpm, spiral reader, measurement every 30 min and 30 °C of temperature for 24 h. Additionally, the filtrate of NB-glycerol was lyophilized to a concentrate in 2X and 5X, which were checked with the same protocol.

### 2.3. Fungal Inhibition Assays

The assay for fungal growth inhibition was carried out in a solid culture of NA-glycerol. Strain L5B5 biomass was spread over agar following a zigzag pattern and incubated at 30 °C for 24 h before the inoculation of fungal species. Once fungi were inoculated, the plates were incubated at 25 °C for 28 days. Fungal species used in this assay were isolated from caves and include *Aspergillus versicolor*, *Penicillium chrysogenum*, *Cladosporium cladosporioides*, *Fusarium solani*, and *Ochroconis lascauxensis* [25,26].

### 2.4. DNA Extraction

Genomic DNA was extracted following two methods, the first one based on enzymatic lysis of cells using the Canvax higher purity™ bacterial genomic DNA isolation kit (Canvax, Cordoba, Spain) for rRNA 16S analysis and short-reads sequencing using Illumina NovaSeq platform, and the second one based on TRIsure™ reagent (Meridian Biosciences, Memphis, TN, USA), for long-reads sequencing with PacBio RSII. The DNA concentration, measured in a Qubit 2.0 fluorometer (Invitrogen, Carlsbad, CA, USA), reached 23.7 ng/μL and 108 ng/μL, respectively.

### 2.5. RNA Extraction

A total of seven samples of biomass from cultures used in the first and second assays were collected for RNA extraction. Four samples from biomass of strain L5B5 cultured with NB supplemented with the carbon sources glycerol, glucose, mannose, and fructose, in the stationary phase of growth, and one from NB with glycerol in death phase. Additionally, two samples were collected from solid cultures of NA with glycerol in exponential and stationary phase. RNA was extracted using TRIsure and Direct-zol RNA Miniprep kit (Zymo Research, Irvine, CA, USA). Concentration was measured in a Qubit 2.0 fluorometer reaching >200 ng/uL for every sample. RNA Integration Number (RIN) was assessed using an Agilent Bioanalyzer 2100 (Agilent, Santa Clara, CA, USA), rating values between 5.5 and 10.

### 2.6. rRNA 16S Analysis

For 16S rRNA amplification, the primers 616F (5′-AGAGTTTGATYMTGGCTCAG-3′) and 1510R (5′-GGTTACCTTGTTACGACTT-3′) were used [27,28]. Polymerase chain reac-

tion was performed in a BioRad iCycler thermal cycler (BioRad, Hercules, CA, USA) using the following cycling parameters: 2 min of initial denaturing step at 95 °C, followed by 35 cycles of denaturing (95 °C for 1 min), annealing (55 °C for 1 min) and extension (72 °C for 2 min), with an additional extension step at 72 °C for 10 min at the end. PCR products were sequenced by Macrogen Inc. (Amsterdam, The Netherlands). The curated and assembled sequence was compared to the EZBioCloud database [29]. The phylogenetic analysis was carried out after 16S rRNA sequences alignment and comparison of strain L5B5 with corresponding sequences of members of the genus *Pseudomonas*. Alignments were created using the multiple sequence alignment program MUSCLE [30]. A phylogenetic tree was created using MEGA X [31]. The evolutionary relationship among bacteria was deduced using the maximum-likelihood method, Tamura-Nei model with a discrete Gamma distribution and invariant sites [32].

### 2.7. Genomic Analysis

DNA extracted for whole genome shotgun was sequenced by Macrogen Inc. (Seoul, Korea) for both Illumina short-reads 150 PE and PacBio long-reads generation, using platforms and library construction as described by Gonzalez-Pimentel et al. [33]. De novo assembly was performed with Canu using the PacBio raw reads [34]. Identification and trimming of overhangs as well as circularization of contigs was carried out with Circlator [35]. Plasmid's identification was checked mapping the Illumina short-reads against the resulted PacBio-assembled contig by means of Burrows-Wheeler aligner and SAM tools [36,37]. Correction and improving of assembly were carried out with Pilon using Illumina short reads [38].

Pairwise genome comparison was carried out using JSpeciesWS web tool. Tetra correlation search (TCS) was deployed to look for closest genomes, which were engaged along with strain L5B5 to calculate the Average Nucleotide Identity with BLAST (ANIb) and MUMmer (ANIm) algorithms. Phylogenomics was implemented with Anvi'o platform concatenating 71 common genes present in analyzed bacteria. The evolutionary relationship among bacteria was deduced using the maximum-likelihood method, JTT model [39] with a discrete Gamma distribution to model evolutionary rate differences among sites [40]. Functional characterization of genes was obtained with Prokka and antiSMASH 5.0 with strict mode and all extra features for automated genome mining of secondary metabolism. Predicted gene clusters included in secondary metabolism were annotated with UniProt database using BLAST. For prediction of virulence factors and resistome, the automated tools Virulence Factors of Pathogenic Bacteria (VFDB) and the Comprehensive Antibiotic Resistance Database (CARD) were used [41,42]. The map of the circular genome was generated with CGView [43].

### 2.8. Transcriptomic Analysis

RNA-based analysis started with the pre-treatment of raw data to finally map the resulted reads over the assembled genome. Default parameters were used for all bioinformatics tools unless otherwise specified. Raw reads were rRNA filtered with SortMeRNA [44]. Once ribosomal RNA was removed, quality trimming and adapter removal were carried out using trimmomatic [45] with the parameters SLIDINGWINDOW:5:20 and MINLEN:75. Filtered reads were aligned to the assembled genome using STAR. First, the genome was indexed with an overhang of 149, and, secondly, reads were aligned to the indexed genome setting the parameters "alignIntronMin" and "alignIntronMax" to 1. Sorted BAM file was indexed with samtools. To count the reads overlapping the genome features, the htseq-count tool in mode "intersection-nonempty" and "reverse" stranded were used. Count normalization and low-expressed gene filtering were carried out by means of NOISeq [46] package from Bioconductor 3.14 [47]. The relationship between culturing conditions was measured with UpSet [48].

## 3. Results

### 3.1. Inhibition Assays

*Pseudomonas* sp., strain L5B5, was selected within a set of airborne mine bacteria after a first pathogenic bacterial and fungal inhibition assays. Strain L5B5 was able to prevent the growth of *Bacillus cereus* and *Staphylococcus aureus*, as well as the inhibition of all the fungal strains used in the test. A second assay for pathogenic bacteria inhibition using glycerol as main source of carbon resulted in a progressive inhibition of pathogens depending on the filtrate concentration. Thus, the crude filtrate had no effect or slight bacteriostatic influence on pathogenic bacteria. The concentrated filtrate 2X inhibited *B. cereus*, whereas the 5X filtrate inhibited *B. cereus*, *S. aureus*, and *A. baumannii* (Figure 1).

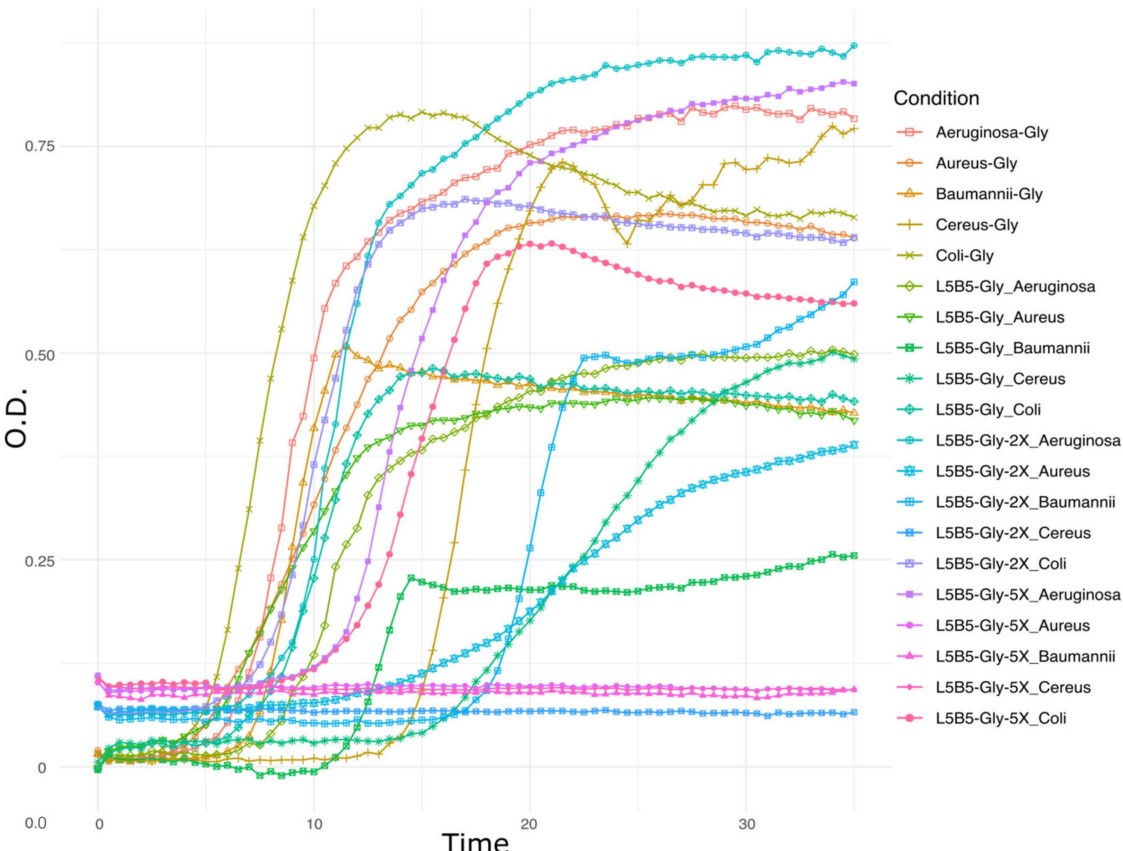

**Figure 1.** Second inhibition assay of pathogens by *Pseudomonas* sp., strain L5B5, filtrates after culturing with glycerol as main source of carbon. Positive control: Aeruginosa-Gly is *Pseudomonas aeruginosa* in NB-Glycerol; Aureus-Gly is *Staphylococcus aureus* in NB-Glycerol; Baumannii-Gly is *Acinetobacter baumannii* in NB-Glycerol; Cereus-Gly is *Bacillus cereus* in NB-Glycerol; Coli-Gly is *Escherichia coli* in NB-Glycerol. Inhibition test: L5B5-Gly is crude filtrate of strain L5B5 culture with NB-Glycerol; L5B5-Gly-2X is 2X concentrated filtrate of strain L5B5 with NB-Glycerol; L5B5-Gly-5X is 5X concentrated filtrate of strain L5B5 with NB-Glycerol.

Likewise, results in a second assay using filtrates after culturing strain L5B5 with NB and glucose, mannose, and fructose showed a similar behavior than when cultured with glycerol (Figure 2). That is, a bacteriostatic effect prevailed either with glycerol or alternative source carbons used for the culture of strain L5B5, being especially successful in delaying the exponential phase in *B. cereus* and *S. aureus* and limiting the duration of this phase significantly.

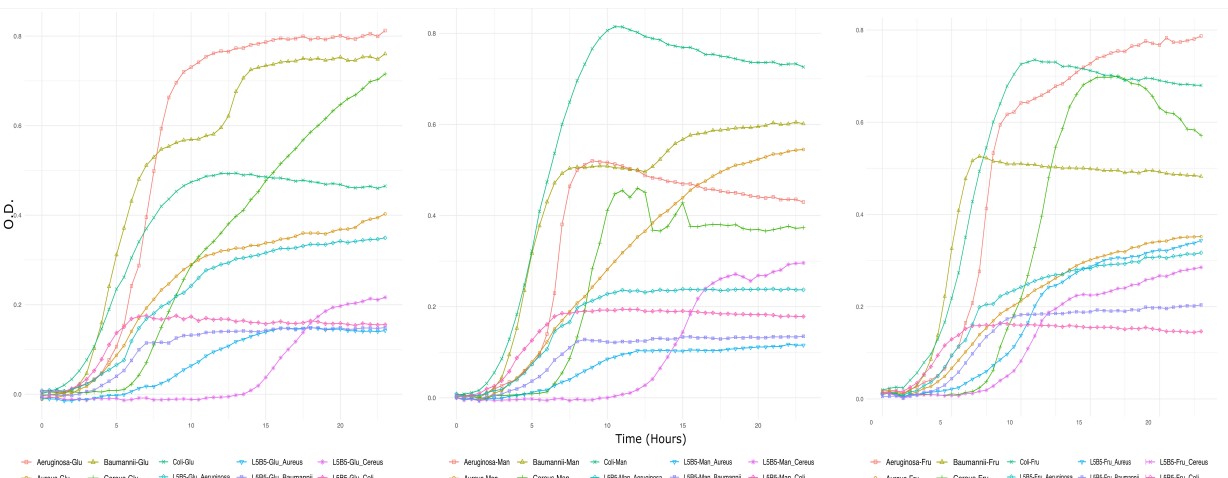

**Figure 2.** Second assay of inhibition of pathogens by *Pseudomona*s sp., strain L5B5, filtrate using alternative source carbons, NB-Glucose (**left**), NB-Mannose (**center**) and NB-Fructose (**right**). Positive controls representing the normal growth of pathogens without the filtrates of *Pseudomonas* sp., strain L5B5 are also displayed in graphics, with the name of the tested species (Aeruginosa, Aureus, Baumannii, Cereus and Coli) and the used carbon sources (Glu-Man-Fru). Inhibition test: Crude filtrates of strain L5B5 culture with NB-Glucose-bacteria are represented as L5B5-Glu-(Aeruginosa, Aureus, Baumannii, Cereus or Coli) (**left**), NB-Mannose-bacteria as L5B5-Mn-(Aeruginosa, Aureus, Baumannii, Cereus or Coli) (**center**), NB-Fructose-bacteria as L5B5-Fru-(Aeruginosa, Aureus, Baumannii, Cereus or Coli) (**right**).

A fungal inhibition assay showed the capability of strain L5B5 to inhibit completely the fungal growth of *Aspergillus versicolor*, *Penicillium chrysogenum*, *Cladosporium cladosporioides*, and *Ochroconis lascauxensis*, whereas in the case of *Fusarium solani* the limitation of growth was relevant (Figure 3A), especially when comparing with the positive control (Figure 3B), where an environmental *Streptomyces* sp. was used to check the growth of fungi with a non-antifungal producer bacterium.

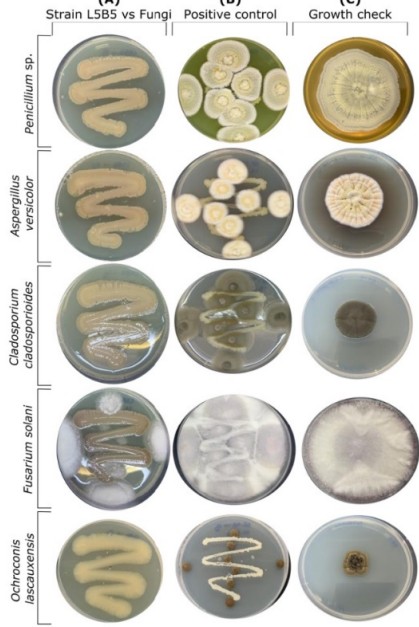

**Figure 3.** *Pseudomonas* sp., strain L5B5, fungal inhibition of growth assay (**A**), in comparison with positive control using an environmental bacterium without antifungals production (**B**) and normal fungal growth in NA-Glycerol plates (**C**).

### 3.2. 16S rRNA Analysis

Based on the 16S rRNA gene sequence similarity, strain L5B5 was affiliated to the genus *Pseudomonas*, being mostly related to *P. piscis* MC042 with 99.20% of similarity, followed of *P. alloputida* Kh7 and *P. juntendi* BML3T with a similarity of 98.69% and 98.62%, respectively. The phylogenetic tree (Figure 4) showed that strain L5B5 was grouped with *P. piscis* MC042, presenting relatedness with a cluster conformed by *P. baetica* a390, *P. laurysulfativorans* AP3 22, *P. mohnii* DSM 18327, *P. moorei* RW10, and *P. rhizosphaerae* DSM 16299.

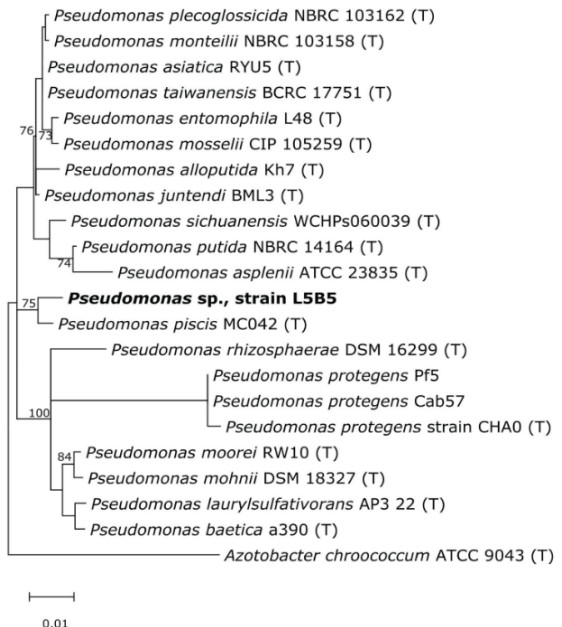

**Figure 4.** Maximum-likelihood phylogenetic tree based on rRNA 16S gene sequences, showing relationship of *Pseudomonas* sp., strain L5B5, with the closest species of the genus *Pseudomonas*. Bootstrap values (>50%) are represented. *Azotobacter chroococcum* ATCC 9043 was used as outgroup.

### 3.3. Genomics Analyses

The whole genome of *Pseudomonas* sp., L5B5 strain, was assembled following a hybrid methodology using Pacific Biosciences long reads and including Illumina short reads 150PE polishing as described by Gonzalez-Pimentel et al. [33].

Genome assembly process resulted in a circular 6,811,662 bp chromosome with a GC content of 63.2% and without presence of plasmids. Genome annotation with prokka resulted in 6123 predicted genes from which 1968 genes were annotated by Sma3s using the curated database UniProt-SwissProt (Figure 5).

Ignoring common and most abundant biological process categories, such as transport and transcription, the number of genes involved in "Stress response", "Virulence", "Antibiotic resistance", "Nitrogen fixation", and "Antibiotic biosynthesis", were 74, 56, 47, 33, and 26, respectively.

For pairwise genome comparison using JSpecies, first, we launched the Tetra Correlation Search tool (TCS) against its own database, showing *P. piscis* MC042, *P. protegens* Cab57, *P. protegens* Pf-5, and *P. protegens* CHA0 to be the closest relatives. Afterwards, these genomes and those from the closest relatives of strain L5B5 observed in the 16S rRNA gene phylogenetic tree were compared to establish the relatedness between bacteria by means of average nucleotide identity using BLAST (ANIb) and MUMmer (ANIm) (Figure 6). The data suggest that *Pseudomonas* sp., L5B5 strain, might represent a new species. Further works for describing the species are ongoing.

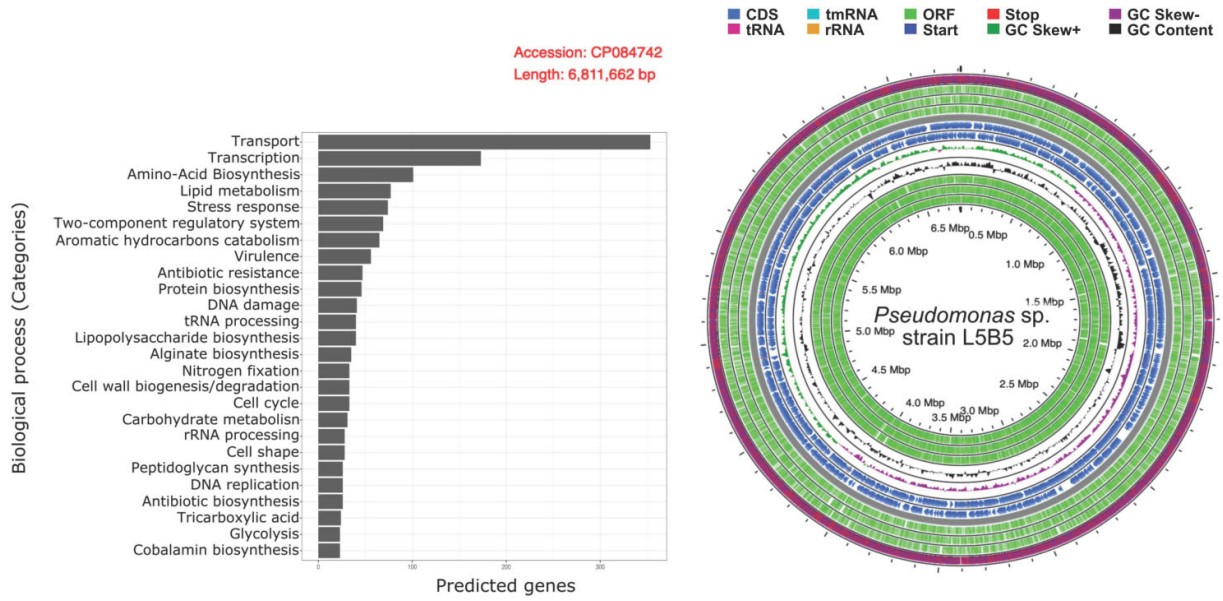

**Figure 5.** Biological processes annotations for *Pseudomonas* sp., strain L5B5 and complete genome representation.

| ANIb / ANIm | 1 | 2 | 3 | 4 | 5 | 6 | 7 | 8 | 9 | 10 |
|---|---|---|---|---|---|---|---|---|---|---|
| 1 | 100 | 89.31 / 90.36 | 86.55 / 88.53 | 86.52 / 88.59 | 86.61 / 88.60 | 79.94 / 85.51 | 76.99 / 84.89 | 79.92 / 85.43 | 79.88 / 85.42 | 79.19 / 85.31 |
| 2 | 89.38 / 90.37 | 100 | 85.93 / 88.10 | 85.90 / 88.19 | 85.96 / 88.17 | 79.97 / 85.31 | 77.22 / 84.93 | 79.91 / 85.39 | 80.00 / 85.37 | 79.08 / 85.16 |
| 3 | 86.41 / 88.54 | 85.52 / 88.10 | 100 | 98.06 / 98.38 | 98.55 / 98.58 | 80.75 / 85.80 | 77.13 / 85.02 | 80.66 / 85.80 | 80.66 / 85.77 | 79.63 / 85.59 |
| 4 | 86.42 / 88.60 | 85.58 / 88.19 | 98.03 / 98.38 | 100 | 98.36 / 98.87 | 80.64 / 85.84 | 83.51 / 85.06 | 80.61 / 85.86 | 80.65 / 85.79 | 79.55 / 85.62 |
| 5 | 86.48 / 88.60 | 85.59 / 88.17 | 98.40 / 98.60 | 98.65 / 98.87 | 100 | 80.75 / 85.84 | 77.11 / 85.11 | 80.60 / 85.88 | 80.68 / 85.79 | 79.68 / 85.59 |
| 6 | 80.01 / 85.51 | 79.78 / 85.37 | 80.83 / 85.80 | 80.81 / 85.84 | 80.84 / 85.84 | 100 | 76.73 / 84.86 | 91.52 / 92.93 | 85.70 / 87.94 | 83.02 / 86.57 |
| 7 | 77.62 / 84.90 | 77.68 / 84.95 | 77.95 / 85.04 | 78.01 / 85.13 | 78.03 / 85.12 | 77.28 / 84.87 | 100 | 77.28 / 84.75 | 77.24 / 84.71 | 77.10 / 84.67 |
| 8 | 79.85 / 85.43 | 79.68 / 85.39 | 80.67 / 85.80 | 80.68 / 85.86 | 80.66 / 85.87 | 91.48 / 92.74 | 76.66 / 84.78 | 100 | 85.35 / 87.97 | 82.81 / 86.62 |
| 9 | 79.82 / 85.42 | 79.77 / 85.36 | 80.70 / 85.77 | 80.69 / 85.78 | 80.78 / 85.78 | 85.41 / 87.92 | 76.48 / 84.70 | 85.12 / 87.95 | 100 | 82.89 / 86.74 |
| 10 | 79.38 / 85.30 | 79.40 / 85.16 | 80.10 / 85.59 | 80.09 / 85.62 | 80.13 / 85.60 | 83.39 / 86.56 | 76.85 / 84.66 | 83.15 / 86.62 | 83.19 / 86.74 | 100 |

**Figure 6.** Pairwise genome comparison of *Pseudomonas* sp., strain L5B5, and closest relatives by means of Average Nucleotide Identity using BLAST (ANIb) and MUMmer (ANIm). 1: *Pseudomonas* sp. strain L5B5; 2: *P. piscis* MC042; 3: *P. protegens* Cab57; 4: *P. protegens* Pf-5; 5: *P. protegens* CHA0; 6: *P. moorei* RW10; 7: *P. rhizosphaerae* DSM 16299; 8: *P. mohnii* DSM 18327; 9: *P. laurylsulfativorans* AP3 22; 10: *P. baetica* a390.

### 3.4. Functional Annotations

Once phylogeny analysis was completed, we proceeded with the secondary metabolism cluster prediction of strain L5B5 based on antiSMASH tool, where *P. piscis* as well as the three strains of *P. protegens* were included due to their tight-knit relationship. Quantitatively, the secondary metabolism was more dynamic in strain L5B5 with the prediction of 20 gene clusters, followed of *P. piscis* with 18, *P. protegens* Pf-5, with 16, and *P. protegens* strains CHA0

and Cab57, with 15 each (Table 1), although it is necessary to consider that antiSMASH divides pyoverdine metabolite in two separated gene clusters. Every predicted gene cluster was blasted against UniProtKB database of UniProt to find a closest approximation for functional annotation. Strain L5B5 shared common secondary metabolites, also called core metabolites, with relatives, such as the mentioned pyoverdine, orfamide biosurfactants, the antimicrobial pyrrolnitrin, fengycin, and 2,4-diacetylphloroglucinol, as well as unknown and unspecified secondary metabolism-like gene clusters.

**Table 1.** Prediction of secondary metabolites in *Pseudomonas* sp., strain L5B5. Similarity of core biosynthetic genes and presence of metabolites in closer relatives are shown.

| *Pseudomonas* sp., strain L5B5 | | 1 | 2 | 3 | 4 |
|---|---|---|---|---|---|
| **Metabolite-Region (R)** | **UniProtKb Similarity (Core Biosynthetic Genes)** | | | | |
| Other-R1 (Pyrrolnitrin) | 95% *Pseudomonas* sp. CMR5c (*prnB*) | + | + | + | + |
| NRPS-R2 (Enantio/Iso-pyochelin) | 90%–95% *Pseudomonas* sp. CMR5c (*pchEF*) | − | + | + | + |
| Phenazine-R3 | 84.7%–85.3%–98.2% *P. piscis*-*P. yamanorum*-*Pseudomonas* sp. FW507-12TSA (*phzI*-*phzAB*) | + | − | − | − |
| T1PKS-R4 (Pyoluteorin) | 84.1%–77% *Pseudomonas* sp. MSSRFD41 (PltBC-like PKSynthase) | − | + | + | + |
| NRPS-R5 (Orfamides) | 89.3%–90.3%–90% *Pseudomonas* sp. CMR5c (*ofaABC*) | + | + | + | + |
| CDPS-R6 | 97.6% *P. protegens* CHA0 (Cyclodipeptide synthase) | + | + | + | + |
| RiPP-like-R7 | 86.3% *Pseudomonas* sp. FW507-12TSA (Uncharacterized protein) | + | + | + | + |
| Arylpolyene-R8 | 89.7%–97.3% *P. protegens* CHA0-*Pseudomonas* sp. CMR5c (Beta-ketoacyl-ACP synthases) | + | + | + | + |
| RiPP-like-R9 | 93.2% *Pseudomonas* sp. CMR12a (Uncharacterized protein) | + | + | + | + |
| T3PKS-R10 (2,4-diacetylphloroglucinol) | 94.6% *P. protegens* Pf-5 (*phlD*) | + | + | + | + |
| Siderophore-R11 | 80.7% *Pseudomonas* sp. CMR12a (Uncharacterized protein) | − | − | − | − |
| Redox-cofactor-R12 (Lankacidin C) | 98.8%–96.7%–97.2% *Pseudomonas* sp. CMR12a-*P. protegens* Pf-5 (*pqqABCDEF*) | + | + | + | + |
| NRPS-R13 (Putative NRPS) | 88.7% *P. juntendi*/*P. putida* (Thioesterase) | − | − | − | − |
| Hserlactone-R14 | 80.5% *Pseudomonas* sp. FW507-12TSA (*psyL*) | + | − | − | − |
| Arylpolyene-R15 | 90.6% *Pseudomonas* sp. CMR5c (Thiolase) | − | − | − | − |
| NAGGN-R16 | 98.7%–96%–97.6% *Pseudomonas* sp. BIOMIG1BAC (M42 family peptidase-N-acetylglutaminylglutamine synthetase | + | + | + | + |
| NRPS-R17-18 (Pyoverdine) | 96.4%–96%–94.3%–95.2% *P. protegens* CHA0 (*pvdLIJD*) | + | + | + | + |
| NRPS-R19 | 85.4% *Pseudomonas* sp. CMR5c (Thioester reductase-like domain) | − | − | − | − |
| Betalactone-R20 | 94.8%–95% *P. piscis*-*Pseudomonas* sp. CMR12a (Acyl-CoA synthase-pyruvate carboxylase) | + | + | + | + |

1: *P. piscis* MC042; 2: *P. protegens* CHA0; 3: *P. protegens* Cab57; 4: *P. protegens* Pf-5. Presence of metabolite in relatives: present (+), not present (−). Metabolite type: Other, not categorized metabolite type; NRPS, Non-Ribosomal Peptide Synthase; T1PKS, Polyketide Synthase Type 1; CDPS, tRNA-dependent Cyclodipeptide Synthases; RiPP-like, unspecified ribosomally synthesized and post-translationally modified peptide product (RiPP); arylpolyene, Aryl polyene; T3PKS, Polyketide Synthase Type 3; hserlactone, Homoserine lactone; NAGGN, N-acetylglutaminylglutamine amide; betalactone, beta-lactone containing protease inhibitor. The most similar known cluster is added in parenthesis.

Genes involved in the synthesis of pyoluteorin and the pyochelin-type siderophores were not observed in *P. piscis*. The production of phenazines were predicted exclusively in strain L5B5 and *P. piscis*. Non-ribosomal peptide synthetase-type clusters 13 and 19, one siderophore in region 11, as well as one homoserine lactone and one Aryl polyene cluster, in region 14 and 15, respectively, were unique in strain L5B5. UniProt similarity using BLAST for strain L5B5 resulted in a major relatedness of genes cluster with undescribed and described *Pseudomonas*, especially with *Pseudomonas* sp. CMR5c and CMR12a as well as with *P. piscis* and *P. protegens* CHA0.

The study of resistome by means of CARD database concluded with the prediction of a major number of genes detected in the strain L5B5 in relation to the closest relatives. A total of seven "Strict" hits were found, two more than those observed in *P. piscis* and

three more than in the three *P. protegens* strains. "Strict" hits identified unknown variants of known antimicrobial resistance mechanisms (ARM) with curated similarity threshold to ensure that detected variants describe a functional antimicrobial gene. All hits found in the analyzed bacteria identified the same Antibiotic Resistance Ontology (ARO) terms, which define the annotation of the ARM [49] with the detection criteria "protein homolog model". The hit ARO:3005040, which describes the interaction of YajC protein with the AcrAB-TolC efflux pump, conferring resistance to certain β-lactam, chloramphenicol, and tetracycline antibiotics, was found in the strain L5B5 and the other related bacteria. The other six hits detected matched with the ARO:3000777, which defines the role of AdeF, the membrane fusion protein of the multidrug efflux complex AdeFGH, to confer resistance to tetracycline and fluoroquinolone antibiotics.

For *Pseudomonas* sp., strain L5B5, the analysis of annotations of virulence factors (VF) with VFDB derived in the prediction of 185 known or potential genes involved in these mechanisms. In *P. piscis*, 233 genes were predicted, whereas 178 were found in *P. protegens* strains CHA0 and Pf-5 and 175 hits in *P. protegens* Cab57. VFDB found homology with described VF in *P. aeruginosa* strains PA7, PAO1 and UCBPP-PA14, *P. fluorescens* strains Pf0-1 and SBW25, *P. protegens* Pf-5, *P. putida* KT2440, *P. syringae* pv. *phaseolicola* 1448A and *P. syringae* pv. *syringae* B728a.

Predicted VF in strain L5B5 included adherence mechanisms, such as flagella, O-specific polysaccharides as major components of lipopolysaccharide (LPS) and type IV pili, antimicrobial activity for phenazines biosynthesis, antiphagocytosis via alginate production, enzyme with hemolytic and non-hemolytic phospholipase C, iron uptake with the already cited pyochelin and pyoverdine, protease through alkaline protease, quorum sensing with the presence of acyl homoserine lactone synthase and N-(3-oxo-hexanoyl)-L-homoserine lactone QS system.

In addition, regulation with GacS/GacA two-component system, secretion system through Hcp secretion island-1 encoded type VI secretion system (H-T6SS) and type III Hop effector proteins, toxin with the presence of exolysin, phytotoxin syringomycin and hydrogen cyanide production, invasion through arylsulfatase activity, lipid and fatty acid metabolism with isocitrate lyase, and stress adaptation by means of catalase were also predicted.

### 3.5. Transcriptomic Analysis—Pre-Treatment and Statistics

Initial pre-treatment for rRNA removing and quality filtering with SortMeRNA and trimmomatic, respectively, conducted to the dropping of few reads, saving between 95.34% of reads for RNA-Seq sample from liquid culture of NB-glycerol in starving phase of growth and 97.62% of the total reads for sample from liquid culture of NB-glycerol in stationary phase of growth (Table 2). Afterwards, filtered reads were mapped against the assembled genome of strain L5B5. STAR statistics showed a percentage of reads mapped between 90.74% and 94.83%, depending on the culturing condition.

**Table 2.** RNA-Seq statistics.

| RNA-Sample | Raw Reads | Reads after rRNA Removing | Filtered Reads | Passed Reads | Mapped Reads |
|---|---|---|---|---|---|
| SGE | 40,589,012 | 40,333,005 | 38,932,275 | 95.92% | 94.46% |
| SGS | 31,304,717 | 31,221,230 | 30,076,663 | 96.08% | 94.17% |
| GYS | 31,931,209 | 31,583,884 | 30,443,705 | 95.34% | 90.74% |
| GLY | 35,637,786 | 35,492,545 | 34,789,606 | 97.62% | 94.06% |
| GLU | 33,422,936 | 32,805,879 | 32,587,198 | 97.50% | 93.38% |
| MAN | 32,739,745 | 32,624,884 | 31,908,361 | 97.46% | 94.83% |
| FRU | 33,195,686 | 33,098,880 | 32,344,406 | 97.44% | 93.90% |

SGE: Solid culture of NA-Glycerol in exponential phase of growth; SGS: Solid culture of NA-Glycerol in stationary phase of growth; GYS: Liquid culture of NB-Glycerol in death phase of growth; GLY: Liquid culture of NB-Glycerol in stationary phase of growth; GLU: Liquid culture of NB-Glucose in stationary phase of growth; MAN: Liquid culture of NB-Mannose in stationary phase of growth; FRU: Liquid culture of NB-Fructose in stationary phase of growth.

The gene expression analysis with NoiSeq led to the normalization of counts performed previously with htseq-count for the different culture conditions by means of trimmed mean

of M-values (TMM) with the filtration of 90% of less expressed genes (622 genes highly expressed genes), to get a minimum of 6080 counts per genome feature or predicted gene. Focusing on condition datasets, liquid cultures in stationary phase of growth had more genes being over this cut-off than observed in other conditions. Thus, the sample from liquid culture with fructose in stationary phase (FRU) was the dataset with more genes overexpressed, whereas the sample from solid culture with glycerol in stationary phase (SGS) was the smaller dataset. The largest set of overexpressed genes was shared among the seven sample's genes independently of the culture conditions and bacterial phases of growth, including 199 genes (Figure 7). Afterwards, liquid culture conditions, including both stationary and death phases, derived in 69 shared features, were followed for samples from liquid culture and stationary phase, when any source carbon was used, equally, with 42 shared genes. To summarize, the variations in culture conditions had a greater incidence in the differential expression of genes analysis, as observed with the solid and liquid cultures, with respect to what was observed when the bacterium phases of growth were evaluated for *Pseudomonas* sp., strain L5B5. On the contrary, the assays using alternative carbon sources showed a low incidence in the differential expression of genes.

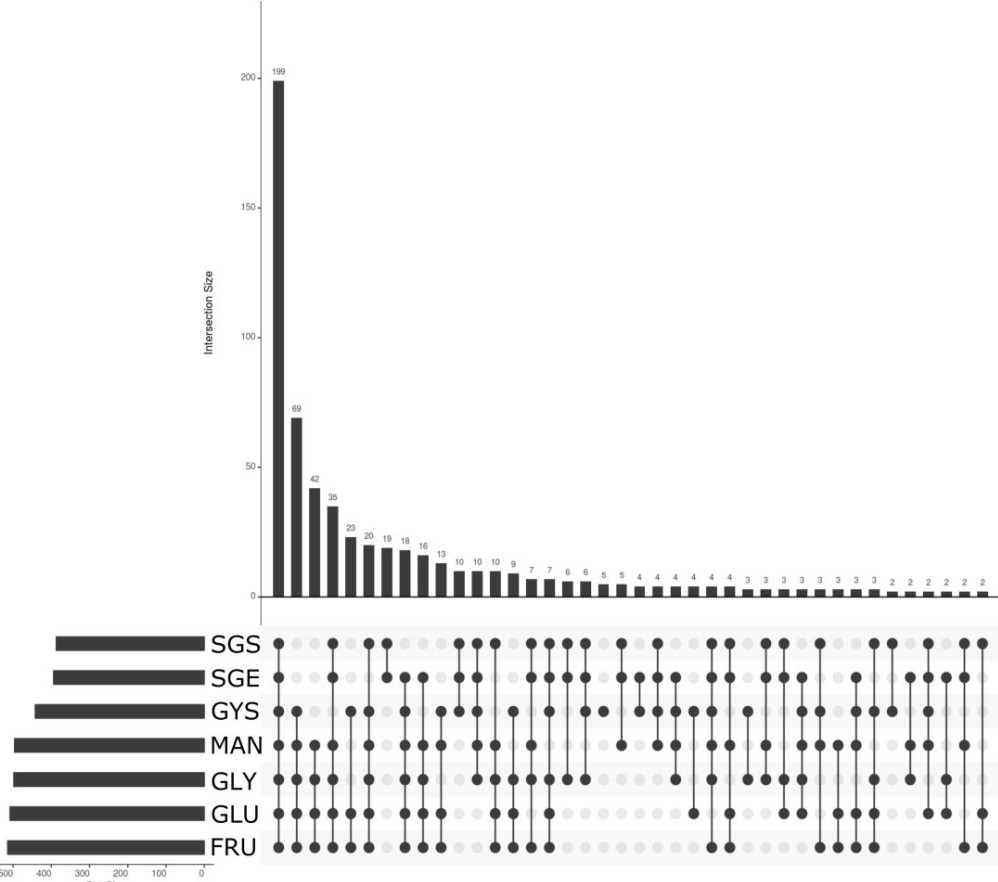

**Figure 7.** Relation of overexpressed genes between conditions for *Pseudomonas* sp., strain L5B5. The figure represents those genes within the 90% of overexpressed genes. SGS: Solid culture. NA-Glycerol. Stationary phase of bacterial growth; SGE: Solid culture. NA-Glycerol. Exponential phase of bacterial growth; GYS: Liquid culture. Nutrient broth with glycerol. Dead phase of growth; GLY: Liquid culture. NB-Glycerol. Stationary phase of growth; GLU: Liquid culture. NB-Glucose. Stationary phase of growth; MAN: Liquid culture. NB-Mannose. Stationary phase of growth; FRU: Liquid culture. NB-Fructose. Stationary phase of growth.

### 3.6. Overexpression in Secondary Metabolism

Within the 90% cut-off, 11 out of the 20 predicted secondary metabolite gene clusters with antiSMASH were identified among the overexpressed genes, observing variations in expression depending on the analyzed condition (Figure 8).

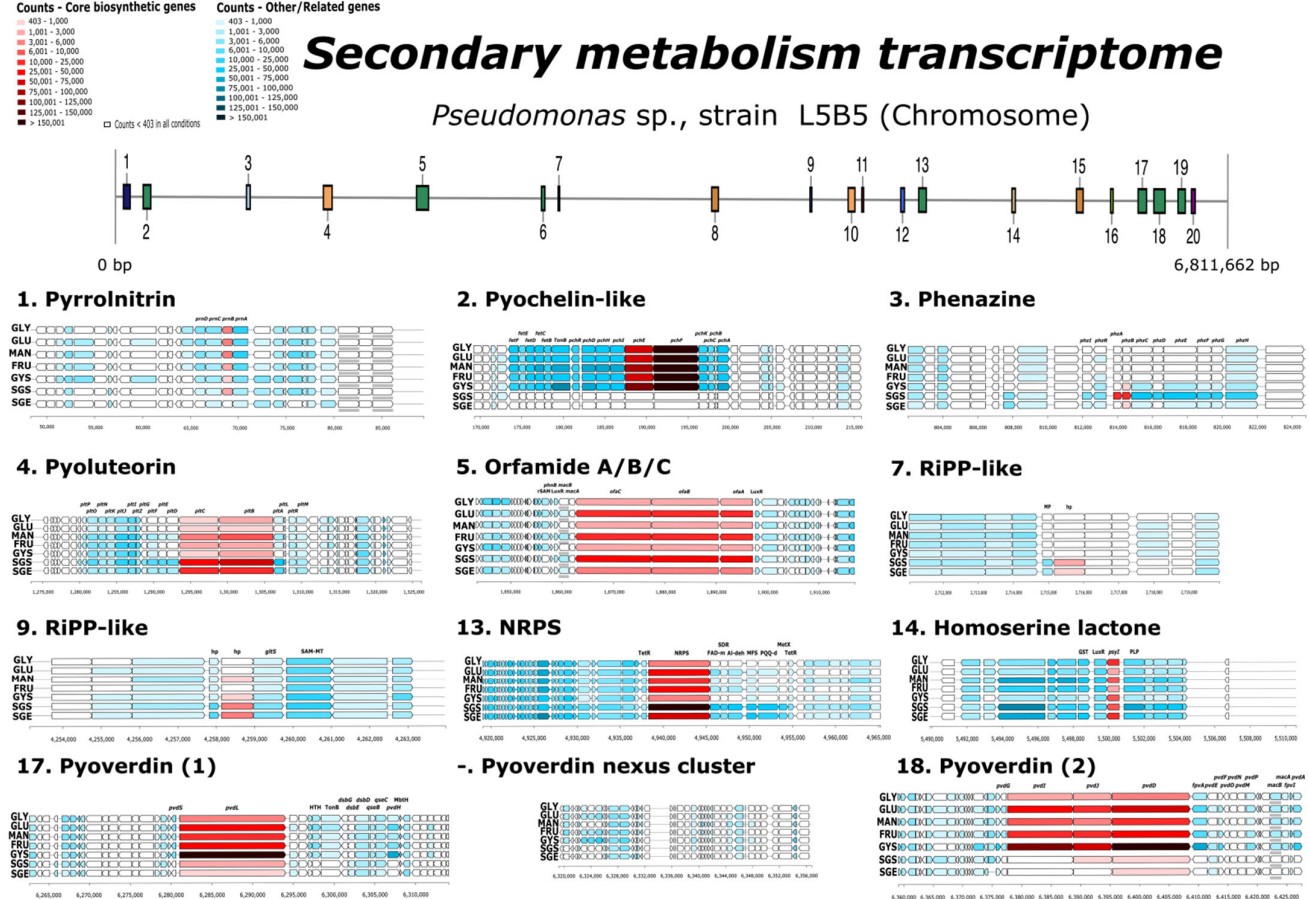

**Figure 8.** Heat map for secondary metabolism cluster predicted for *Pseudomonas* sp., strain L5B5 and overexpression of genes through RNA-seq analyses. GLY: RNA extracted from liquid culture (NB) supplemented with glycerol in stationary phase; GLU: RNA extracted from liquid culture (NB) supplemented with glucose in stationary phase; MAN: RNA extracted from liquid culture (NB) supplemented with mannose in stationary phase; FRU: RNA extracted from liquid culture (NB) supplemented with fructose in stationary phase; SGS: RNA extracted from solid culture (NA) supplemented with glycerol in stationary phase; SGE: RNA extracted from solid culture (NA) supplemented with glycerol in exponential phase.

In region 1, where the closet predicted metabolite was pyrrolnitrin, the overexpression of the described gene cluster prnABCD was observed [22]. Pyrrolnitrin is a phenylpyrrole derivative, with antifungal activity, originally isolated from *Pseudomonas pyrrocinia*, but found in other proteobacteria [50,51]. The presence of glycerol, glucose, and mannose in liquid culture was determinant for an increased gene expression in stationary phase (GLY, GLU, MAN), showing less determination for FRU and SGS. Under conditions of liquid culture with glycerol in death phase of growth (GYS) and solid culture with glycerol in exponential phase (SGE), the gene expression was low or irrelevant.

In region 2, the core cluster pchR-pchDHIEFKCBA for enantio/iso-pyochelin, as well as the protein-dependent ABC transporter encoded by the fetABCDEF operon, were significantly overexpressed in liquid cultures, and particularly in GYS and MAN. During these conditions, the overexpression of the determinant genes for enantio/iso-pyochelin synthesis, fetA and pchEF [21,52], stood out among the related genes.

Genes of the predicted cluster in region 3, which likely encodes some phenazine-type compound, were differentially expressed under SGS condition, with minor or irrelevant expression in the rest ones. The relation of expressed genes determined the operon structures present in strain L5B5, with the core cluster phzABCDEFG [53], the transcriptional regulators via quorum sensing phzRI [54] and the transamidase phzH, which enable the bacteria containing this mechanism to produce both phenazine-1-carboxylic acid (PCA) and phenazine-1-carboxamide (PCN) [55].

Region 4, with antifungal pyoluteorin as the closest metabolite identified, was observed to be overexpressed for every analyzed condition, but more particularly in the two solid culture conditions. In this dataset, the two pair of operons involved in synthesis and transportation of pyoluteorin, pltLABCDEFG, and pltRM, as well as pltPIJKNO and pltZ operons, respectively, were differentially expressed [56,57].

Orfamide-type cyclic lipopeptides (CLPs), identified as the closest metabolite likely produced by gene cluster in region 5, are biosurfactants synthesized by *Pseudomonas* with insecticidal activity and involved in the biocontrol of plant pathogens. Up to eight different orfamides has been described so far, Orfamide A-H, in *Pseudomonas protegens* strains Pf-5 and CHA0, as well as in *Pseudomonas* sp. strains CMR5c and CMR12a [19,58,59]. Cluster prediction derived in the identification of auxiliary genes described in *P. protegens* Pf-5, such as RND efflux system upstream of the core biosynthetic genes ofaABC, and a putative glyoxalase family protein (phnB-like) and a heme transporter CcmD, radical SAM domain-containing protein (rSAM), downstream of two transporter genes (macA and macB) and luxR type transcriptional regulator gene. Core biosynthetic genes were overexpressed in all conditions analyzed, especially in GLU, FRU, and SGS, where auxiliary genes upstream and downstream were highly expressed as well. These conditions showed a higher expression of macB gene compared to macA, suggesting the importance of macB and the downstream regulator in orfamide synthesis, as observed by Olorunleke et al. [60] where in mutants with deletion of LuxR2, the expression of macB was weaker than macA, avoiding the production of orfamides.

Regions 7 and 9 were identified as RiPP-like metabolites, according to antiSMASH "Other unspecified ribosomally synthesized and post-translationally modified peptide product (RiPP) cluster". This means that these gene clusters might induce the synthesis of a secondary metabolite, but none of the described compounds can be assigned to these genetic mechanisms. Genome mining and the recent discovery of RiPPs has awakened interest [61]. Solid culture in both phases of growth was the condition that differentially expressed the suggested core genes, which were identified as "hypothetical protein" in both gene clusters.

A possible NRPS-like metabolite predicted in region 13 didn't match with any described metabolite. The core gene appeared to be differentially expressed under SGS condition and, although it was overexpressed in other conditions, only in this dataset was the overexpression of contiguous genes observed, with the presence of two transcriptional regulator tetR-like genes upstream and downstream of core biosynthetic gene and pyrroloquinoline quinone-like enzyme (PQQ), synthesized by only a few bacteria and with possible relevance in plant growth-promoting bacteria [62]. Though there was no connection of this gene cluster with any described biosynthetic processes in the closest sequences from *P. juntendi*, *P. putida*, and *P. fuscovaginae*, ranging from 88.7% to 81.2% of similarity, the core gene was shown to be certainly related with siderophore biosynthesis from *Pseudomonas* sp. R4-34-07, a plant-beneficial bacterium [63], with 63% of similarity. Other, more distant relatives were *Vibrio spartinae*, isolated from salt-marsh plant [64], and several strains of the species *Xenorhabdus bovienii*, including the entomopathogenic strain SS-2004 [65].

The homoserine lactone gene cluster, found in region 14, has no described metabolite associated, but as a well-known auto-inducer for Gram-negative bacteria it could have function in the production of antibiotics, pathogenicity, virulence, pigment formation, or sporulation [66]. The core gene was overexpressed in every analyzed condition, with high incidence in GLY and SGE.

Finally, two separated gene clusters involved, apparently, in the pyoverdine production, with a self-made nexus gene cluster, not predicted by antiSMASH, were in regions 17 and 18, where core biosynthetic genes were identified as pvdL and pvdIJD, respectively. Pyoverdine is an essential metabolite produced by the named "Fluorescent Pseudomonads" bacteria, where *P. aeruginosa*, *P. fluorescens*, *P. putida* and *P. syringae* are the most important species [67]. This siderophore biogenesis begins after pvdL, pvdI, and pvdD expression, depending on the strain [68–71]. Lacking iron in liquid cultured conditions showed a substantial rising of expression of mentioned and auxiliary genes, being differentially expressed in GYS dataset. Within this condition, pvdL and pvdD were the core genes significantly overexpressed, which are part of the "siderosome", the enzymatic complex located in cytoplasm where the acylated ferribactin precursor of pyoverdine is synthesized. Once this metabolite is generated, it is exported to the periplasm through PvdE protein and deacylated and oxidized to dihydropyoverdine by PvdQ (out of pyoverdine biosynthesis gene cluster) and PvdP, respectively. PvdP is responsible for the complete fluorophore formation [72] and along with PvdO they are the enzymes involved in complete oxidation of dihydropyoverdine. In fact, PvdO is the cause of the final oxidation, without which pyoverdine formation is not possible [73]. Our RNA-Seq analyses resulted in the low expression of pvdO as well as the other two genes that make up the operon pvdMNO, which appear to be essential in pyoverdine formation [73]. Besides, the importation of pyoverdines seemed to be relevant due to the fpvA overexpression, introducing iron in cells. Thus, the overexpression of this gene could be limiting the formation of pyoverdine, as proposed by Lamont et al. [74].

The expression of predicted virulence factors in strain L5B5 with VFDB was assessed among analyzed conditions through a heat map to determine the differential expression of the related genes (Figure 9). Samples from solid cultured were shown to be a differential factor for flagellin expression, involved in flagella VF, and with little emphasis for liquid cultures in stationary phase of growth. SGS was determinant for phenazines, as described before, and although it appeared not to have incidence in siderophores pyochelin/pyoverdine formation, one gene is susceptible to be involved in the synthesis of pyoverdine. This gene was identified as the core for biosynthesis of metabolite in predicted cluster 13 by antiSMASH. The other gene was included as a pyoverdine feature for SGS, which is not comprised in this siderophore gene cluster, and seems to be expressed to a fatty acyl-AMP ligase (FAAL). This enzyme belongs to the class I adenylate, and it has been suggested that the acyl adenylates participate as substrates in the synthesis of complex lipids, such as mycobactin siderophore [75], but they are also found at the N terminus of non-ribosomal peptide synthetases, which catalyze the synthesis of relevant peptide products [76]. SGS and SGE resulted to determine the expression of type VI secretion system (T6SS) related genes. T6SS is a bacteriophage contractile sheath-like structure used by members of Gram-negative bacteria to deliver substrates directly inside a prokaryotic and/or eukaryotic cell. It was originally identified in *P. aeruginosa* and *Vibrio cholerae*, reported to contribute to virulence in some pathogenic bacteria [77].

GYS condition appears as the main cause for expression of related VF in strain L5B5. Excluding the flagellin, GYS showed to be important in flagella synthesis, as well as for alginate regulation. Alginate takes a relevant role in *P. aeruginosa* and *P. syringae* acting as a main virulence factor, but also the synthesis of this product serves to cells as source of carbon [78,79]. However, where GYS proved determinant is in the genesis of siderophores, especially pyoverdine. Conditions FRU and GLU behaved as differential condition to alkaline protease VF expression. Alkaline protease is involved in virulence mechanisms in *P. aeruginosa* and seems to contribute to pyocyanin production [80].

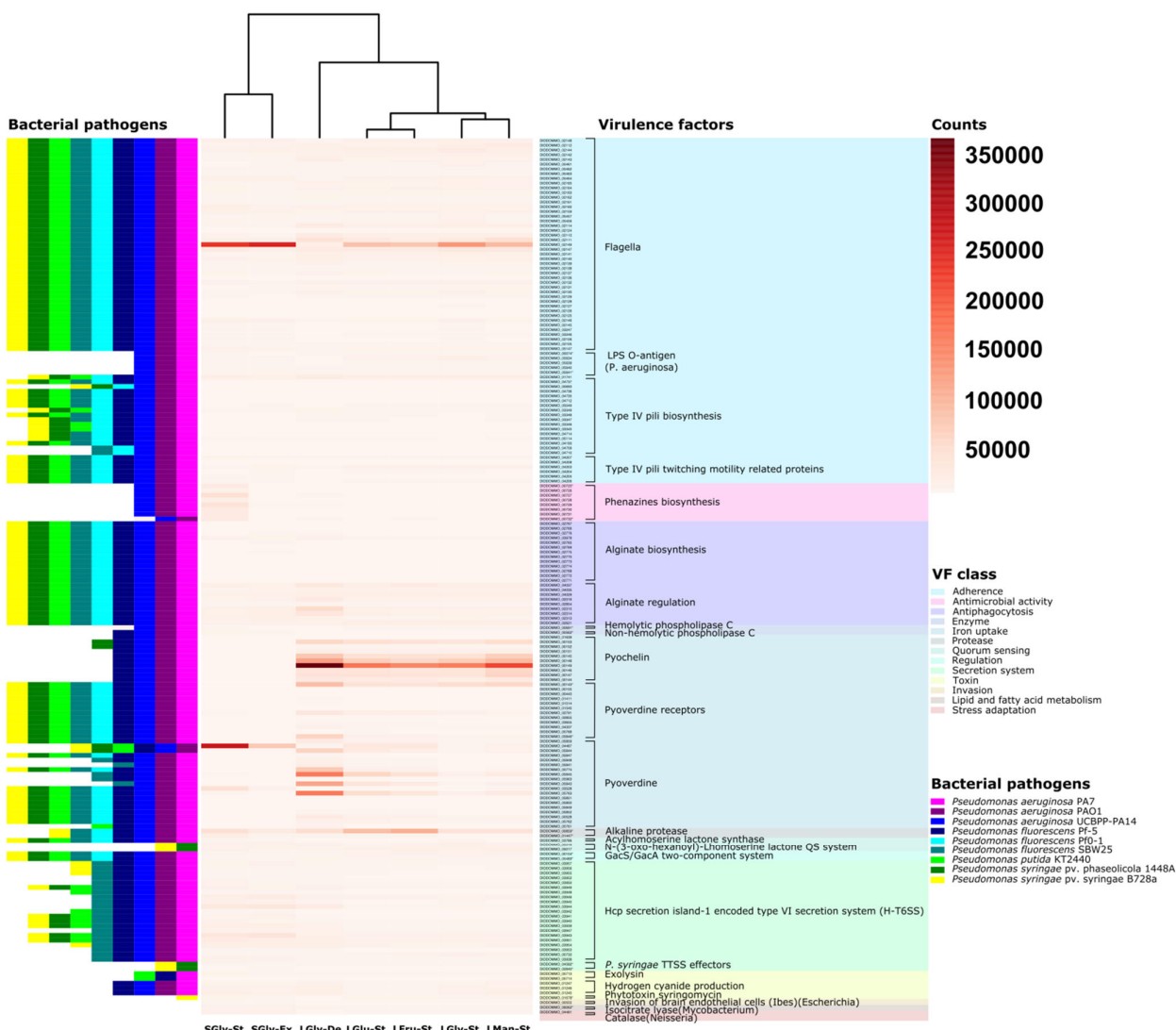

**Figure 9.** Heat map for predicted and expressed virulence genes in *Pseudomonas* sp., strain L5B5. Prediction of genes involved in virulence factors (**right**), and relation of bacterial pathogens (**left**) which own the same factors, according to the VFDB database.

## 4. Discussion

The preservation of industrial heritage sites for research and educational visits is a recent ethno-touristic activity. These include inactive mines, which are reconditioned, aiming to exhibit and make aware of old mining procedures to extract minerals and how in former times they affected the social and economic life in the region. We explored the old Lousal mine, open to visits, for aerobiological studies.

The origin and characterization of airborne microorganisms in subsurface environments represent an important research topic when public access is in consideration. The exclusive features of caves and mines determine the presence of microorganisms inside, acting as selective factor and leading to the adaptation of their metabolism. The isolation of *Pseudomonas* sp., strain L5B5, and its characterization using both in vitro and in silico analyses delve into the metabolic mechanisms and how this bacterium could interact with the environment.

Phylogenetic analyses included strain L5B5 in the fluorescent pseudomonads group, where diverse species, such as the opportunistic pathogen *P. aeruginosa* and the growth-promoting plant *P. protegens*, are included. rRNA 16S gene sequence related strain L5B5

with valid species, such as *P. alloputida*, isolated from rhizosphere of bean plant in Iran [81], *P. juntendi*, a bacterium isolated from sample from a sputum sample of a patient in Japan [82] and the closest relative, *P. piscis,* isolated from the head ulcers of farmed Murray cod in China [83], respectively. However, 16S gene phylogenetic tree grouped strain L5B5 with *P. piscis* in the same branch, while *P. alloputida* and *P. juntendi* were located in a distant branch. This result was endorsed with pairwise genome comparison and phylogenomics, where also three strains of *P. protegens* CHA0, Cab57 and Pf-5 presented a close relatedness. Therefore, strain L5B5 seemed to share competences with both opportunistic pathogens and growth-promoting pseudomonads.

The inhibition of *Bacillus cereus, Staphylococcus aureus,* and *Acinetobacter baumanii* in solid cultures with concentrated filtrates from strain L5B5 was remarkable. Recent studies reported the interaction of *P. aeruginosa* with *S. aureus*, stimulating the vancomycin susceptibility, and *A. baumannii*, PCN mediating [84,85]. Strain L5B5 had a high effectivity against all tested fungi, originally isolated from subsurface environments. The presence of fungi in caves and mines is directly related with airborne transport, terrestrial environments, including pathogens and plant endophytic species, and insects [26,86,87].

Functional annotations from the assembled genome of strain L5B5 showed a significant abundance of genes involved in biological processes such as stress response, virulence, antibiotic resistance, or nitrogen fixation that could also be linked to similar behaviors in relatives. The prediction of secondary metabolites and gene similarities led to the detection of common mechanisms found in the biocontrol of bacteria, such as the specific antimicrobials pyrrolnitrin, pyoluteorin, and phenazines, as well as the combination of phenazines with the cyclic lipopeptides orfamides, as reported D'aes et al. [58]. The expression of genes involved in the synthesis of these compounds in SGS condition should be definitive to achieve the inhibition of the tested fungi since the activity of these metabolites have been verified separately [22,88,89]. However, the absence of phenazines in liquid-cultured conditions placed siderophores as the most likely factor to inhibit the growth of *S. aureus* and *B. cereus*, since *E. coli* has been probed to share in common the same binding site that *P. aeruginosa*, with no affection of raw or concentrated filtrates as well [90,91], leading to the recognition of iron-free siderophores and, in this way, its growth is not affected when concentrated filtrate was added to the in vitro second assay analysis. In a co-culture with *P. aeruginosa*, the viability of *S. aureus* was reduced after siderophores and quorum sensing exoproducts synthesis as described mechanisms against Gram-positive bacteria [92]. Inhibition of *A. baumannii* is unexplained when common siderophores synthesis is concerned since it has been demonstrated the high similarity between the iron receptors structures or TonB-dependent receptors in *P. aeruginosa* and *A. baumannii*, which should confer a similar behavior to that observed in *E. coli* [93]. Phenazines should not be the cause of the inhibition of *A. baumannii*, since the expression of the genes associated with their production was negligible in liquid cultures, and in the first trial, where biomass from solid culture was used, with parallel overexpression of the genes, the inhibition of this pathogen did not occur. Thus, considering that the differential metabolites produced by strain L5B5 in liquid cultures seemed to be pyoverdine and pyochelin, it is plausible to believe that these compounds could achieve a selective behavior within Gram-negative bacteria.

The non-ribosomal peptide synthetase gene cluster predicted as cluster 13 by antiSMASH sets out a difficult scenario for the identification, or role, of the synthesized compound, since despite some closed sequences, approximately 63% of similarity was shown to be related with siderophores, as well as the VF prediction that included the core gene as part of siderophore mechanism. Both pyochelin and pyoverdine, the most common strategies to capture iron, were almost completely produced during liquid culture condition, especially in the death phase of growth. This situation contrasted with the differential expression of the hypothetical gene cluster 13 in SGS condition, where it would be expected to find no iron chelation needs. A few bacteria with closest related sequences were isolated from patients or involved in plant pathogenesis, such as *P. juntendi* or *P. fuscovaginae* [94]. However, pyochelin and pyoverdine siderophores identified in strain

L5B5 have sequences closer to those described in biocontrol agents. Therefore, additional studies should be performed to discover the functionality of cluster 13 in this bacterium.

## 5. Conclusions

Here, we report the in vitro and in silico analyses of *Pseudomonas* sp., strain L5B5, isolated from the air of a public-accessed gallery from the pyrite mine of Lousal, Portugal. We assembled the complete genome for functional annotations and transcripts mapping using RNA-seq reads based on different culturing conditions. In this way, we observed the differential expression of genes involved in secondary metabolites, including a feasible new biosynthetic gene cluster with low similarity with previously described sequences in databases. The possible incidence of the airborne *Pseudomonas* sp., strain L5B5, in human health is unknown, but the prediction of mechanisms associated to the resistome, with more hits detected than in the closest relatives, as well as a high presence of genes with virulent potential, is remarkable. This contributes to the identification of possible threats in a mine adapted for visits and public use. A selective and specific study based on these findings could discover new biological processes in microorganisms from mines, as recently described for cave bacteria [95,96].

**Author Contributions:** Investigation: J.L.G.-P., I.D.-M., V.J. and A.T.C. Writing—review & editing: J.L.G.-P. and C.S.-J. Project administration and funding acquisition: C.S.-J. All authors have read and agreed to the published version of the manuscript.

**Funding:** The research was supported by the European Union's project 0483_PROBIOMA_5_E, co-financed by the European Regional Development Fund within the framework of the Interreg V-A Spain-Portugal program (POCTEP) 2014–2020.

**Institutional Review Board Statement:** Not applicable.

**Informed Consent Statement:** Not applicable.

**Data Availability Statement:** The 16S rRNA gene sequence was deposited into the DDBJ/ENA/GenBank database under accession number OK655762. Likewise, the whole-genome sequence for *Pseudomonas* sp., strain L5B5, was deposited in the same database under the accession number CP084742. RNA-Seq sequences and counts were deposited at the Gene Expression Omnibus (GEO) under the accession number GSE186946.

**Acknowledgments:** The authors acknowledge to CSIC Open Access Publication Support Initiative through its Unit of Information Resources for Research (URICI), and CSIC Interdisciplinary Thematic Platform Open Heritage: Research and Society (PTI-PAIS) for the professional support.

**Conflicts of Interest:** The authors declare no conflict of interest.

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
