# Peer review of "Pseudomonas sp., Strain L5B5: A Genomic and Transcriptomic Insight into an Airborne Mine Bacterium"

_applsci, doi:10.3390/app122110854_

Round 1

Reviewer 1 Report

The following manuscript by Gonzalez-Pimentel et al., summitted for publication, looked to characterize a particular Pseudomonas sp., strain found in the air of a retired zinc/copper mine in Portugal. Since mines harbor bacteria with unique ability to resist high concentration of heavy metals and represent complex ecosystems where multitudes of microorganisms cohabit together and form complex intertwined life cycles, including that of known pathogens, the authors decided to characterize in-depth the isolated Pseudomonas sp., strain L5B5, partly for Pseudomonas propensity for being a pathogen, but also for the bacteria’s ability to interact beneficially with other microbes.  Additionally, the authors highlight that studies of airborne bacteria in mine caves are lacking.

The authors conducted an extensive in vitro and in silico analysis of L5B5. This included finding -via coculture or isolated cell filtrate- that this particular strain had a remarkable ability to repress the growth of common bacteria as Bacillus cereus and Staphylococcus aureus, along with multiple types of fungus that inhabit the caves.  Genomic and transcriptomic analysis predicted a reservoir of genes involved in the production of secondary metabolites, possibly contributing to its ability to repress the growth of other microorganisms, along with clusters of virulence genes that could pose a future health threat.

Overall, the manuscript is well written and represents a tremendous amount of work and contributes to our greater knowledge of microorganisms inhabiting abandon/retired mines. Apart from a few minor revisions (see below) I would recommend this manuscript for publication.

Minor Revisions

1.     Pg 5. Line 227 – change “Thus, the crude extract had not effect or slight..” to Thus, the crude extract had no effect or slight…”

2.     Pg10. Line356 to 367 – Please break into smaller sentences.  Currently the sentence is very long and hard to follow.

3.     Pg12. Line 399 – Change “…was shared among the seven samples genes independently..” to “…was shared among the seven sample’s genes independently..”

4.     Pg12. Line 400-403- Please reword the sentence; it’s very hard to follow.

Author Response

All the comments have been considered in the revised manuscript

Reviewer 2 Report

Dear Editor,

I send you here my comments for the manuscript review.

Title: Pseudomonas sp., strain L5B5: A genomic and transcriptomic insight into an airborne mine bacterium.

Suggestions

Introduction:

Line 69: Put “pseudomonas” in italics. Same remark in all the main text.

Line78: Is it "pseudomonads" or "pseudomonas". Please correct if this is a mistake. Same remark in all the main text.

Line 81: The objective of the study is unclear.

Material and methods:

Line 104: Replace “28°C” by “28 °C”. Same remark in all the main text for °C unit.

Results:

Figure 2 caption is unclear to read the content. Please improve it. Same remark for all figures, which are not clear.

Discussion:

The results are well discussed.

I suggest to add a conclusion.

References:

Please check references (in text and list) in relation to the journal's recommendations.

Author Response

(The authors gave the same response as above.)
